# Sexual Dimorphism in Metabolic Responses to Western Diet in *Drosophila melanogaster*

**DOI:** 10.3390/biom12010033

**Published:** 2021-12-27

**Authors:** Sofie De Groef, Tom Wilms, Séverine Balmand, Federica Calevro, Patrick Callaerts

**Affiliations:** 1Laboratory of Behavioral and Developmental Genetics, Department of Human Genetics, KU Leuven, University of Leuven, B-3000 Leuven, Belgium; tom.wilms@kuleuven.be; 2Université Lyon, INRAE, INSA Lyon, BF2I, UMR 203, 69621 Villeurbanne, France; severine.balmand@inrae.fr (S.B.); federica.calevro@insa-lyon.fr (F.C.)

**Keywords:** *Drosophila melanogaster*, obesogenic diets, western diet, metabolism, sexual dimorphism

## Abstract

Obesity is a chronic disease affecting millions of people worldwide. The fruit fly (*Drosophila melanogaster*) is an interesting research model to study metabolic and transcriptomic responses to obesogenic diets. However, the sex-specific differences in these responses are still understudied and perhaps underestimated. In this study, we exposed adult male and female *Dahomey* fruit flies to a standard diet supplemented with sugar, fat, or a combination of both. The exposure to a diet supplemented with 10% sugar and 10% fat efficiently induced an increase in the lipid content in flies, a hallmark for obesity. This increase in lipid content was more prominent in males, while females displayed significant changes in glycogen content. A strong effect of the diets on the ovarian size and number of ma-ture oocytes was also present in females exposed to diets supplemented with fat and a combina-tion of fat and sugar. In both males and females, fat body morphology changed and was associ-ated with an increase in lipid content of fat cells in response to the diets. The expression of me-tabolism-related genes also displayed a strong sexually dimorphic response under normal condi-tions and in response to sugar and/or fat-supplemented diets. Here, we show that the exposure of adult fruit flies to an obesogenic diet containing both sugar and fat allowed studying sexual dimorphism in metabolism and the expression of genes regulating metabolism.

## 1. Introduction

Obesity is a chronic disease associated with a major socio-economic health impact. Almost two billion adults worldwide are overweight, of whom 650 million are diagnosed as obese [1]. Overall, the prevalence of obesity is higher in females compared to in males. However, obesity-associated disorders such as diabetes and cardiovascular disease are often more prevalent in men. Despite these disparities, the inclusion of both sexes is often lacking in fundamental metabolic research [2]. Unraveling the molecular basis of sex differences in metabolism will likely be relevant to help reduce the numbers of patients with obesity and provide better treatments. Despite physiological differences in development and metabolism, studies in *Drosophila melanogaster* have contributed significantly to our understanding of molecular and genetic mechanisms in metabolic regulation in mammals, including humans [3]. *Drosophila* is an excellent model to study molecular mechanisms mediating metabolic homeostasis, since molecular pathways and neuroendocrine systems that govern energy balance in *Drosophila* are strongly conserved in mammals [4]. *Drosophila* develops obesity and its associated complications in a similar fashion as humans. For example, diet-induced obesity in *Drosophila* leads to fat accumulation in the form of increased triglyceride (TAG) content and lipid droplet number and size [5,6]. The administration of a diet supplemented with either high sugar or high fat concentrations increases larval body weight and insulin resistance and decreases motor and heart functions [6,7,8,9]. Many studies using obesogenic diets investigate larvae or adult flies reared on high-sugar diets (HSDs) [5,6,10,11] or high-fat diets (HFDs) [12], which may bias the responses in adult flies, and possibly worsen certain metabolic parameters [13]. Other studies use adult flies reared under standard conditions and expose them for a specific amount of time to HSDs [7,14] or HFDs [8,15,16,17,18]. Few studies expose flies to western diets, combining both sugar and fat and even salt [19]. Moreover, most obesogenic diets are often supplemented with very high concentrations of sugar (30%) [6,7,20,21] or fat (20–30%) [8,9,22]. The addition of a high concentration of sugar in itself can pose an osmotic challenge and affect survival [14], so that evaluating the metabolic effect of sugar addition becomes difficult. We hypothesized that a modest increase in both fat and sugar is more representative of modern western diets that at least partially underlies a number of human conditions [23]. The differences in obesity incidence in males and females suggest that metabolic homeostasis in response to western diets is sexually dimorphic. Therefore, we aim to characterize sexually dimorphic responses to diets supplemented with both fat and sugar using *Drosophila* as a model system. Adult female and male flies were exposed to a standard diet or a diet supplemented with 10% sucrose, 10% or 20% coconut oil, or a combination of both. A western diet containing 10% sucrose and 10% coconut oil efficiently increased TAG levels, primarily in male flies. Diets supplemented with sucrose and/or coconut oil induced sexually dimorphic responses in metabolism and metabolic gene expression, suggesting a sex-specific regulation of metabolism in response to obesogenic diets.

## 2. Materials and Methods

### 2.1. Flystock and Flyrearing

Wild-type strain *white*^Dahomey^ *Drosophila melanogaster* (a kind gift of Carlos Ribeiro) was challenged with different dietary conditions and was used for the evaluation of metabolic parameters. *Dahomey* flies were reared on Nutri-Fly^®^ Bloomington Formulation fly food (product # 66-112; Genesee Scientific, San Diego, CA, USA) at room temperature (±21 °C). Nutri-Fly food was prepared from 180 g of powder batches composed of 9.61% yeast, 5.55% soy flour, 40.6% yellow cornmeal, 3.20% agar, and 42.7% light corn syrup. Each batch was dissolved in 1 L of demineralized water and heated until boiling under constant stirring. Then, 4.8 mL of propionic acid was added to the mixture prior to being dispensed into vials. Following eclosion, the flies were aged for 3 days, after which male and female flies were transferred at a 1:1 number ratio to the diets, which consisted of Nutri-Fly food supplemented with varying concentrations of coconut oil (*v*/*v*%: 0%, 10%, and 20%; Acros organics 365475000, Geel, Belgium) and/or sucrose (*w*/*v*%: 0%, 5%, 10%, 20%, and 30%; product #S0389, 1 kg; Sigma-Aldrich, St. Louis, MO, USA) (Table 1). Diets were prepared by mixing Nutrifly powder with 10% sucrose and/or 10% or 20% coconut oil. The mixture was added to a beaker with water and cooked according to the manufacturer’s instructions. The mixture was allowed to reach a soft boil, followed by a 10 min simmer, until the food became more viscous, after which the heat was turned down and the food was dispensed into vials. The food was allowed to set and cool to room temperature. The food was cooked fresh in small batches. No fungal or microbial growth was observed in any of the diets. Upon the addition of the flies, all vials were kept in a horizontal position to prevent flies from sticking into the food. A total of 15 males and 15 females were kept per vial to assure controlled age and density. The flies were maintained on the diets for 7 days at room temperature (20–21 °C) to prevent food enriched with fat from melting. The flies were transferred once to a fresh vial during the 7-day experiment, to avoid the accumulation of feces and eggs.

### 2.2. TAG Measurement

TAG measurement was performed according to a slightly modified protocol based on [24,25]. After 7 days on the diets, the flies were anaesthetized with CO_2_ gas. A total of 5 adult flies per replicate with 4 replicates per condition were collected. Two hundred and twenty microliters PBST (1X PBS + 0.05% Tween 20) were added to each sample. The samples were mechanically homogenized in PBST on ice using pestles and a motorfollowed by inactivation by heating to 70 °C for 10 min. Next, the samples, a TAG reagent (product #981786; Thermo Fisher Scientific, Waltham, MA, USA), and a free glycerol reagent (product #F6428; Sigma-Aldrich, St. Louis, MO, USA) were warmed to 37 °C. Meanwhile, a set of free glycerol standards was prepared in PBST (1.0 mg/mL, 0.80 mg/mL, 0.6 mg/mL, 0.5 mg/mL, 0.25 mg/mL, 0.125 mg/mL, and 0 mg/mL) from a 2.5 mg/mL stock (product #G7793; Sigma-Aldrich, St. Louis, MO, USA). Then, 25 µL of each standard or sample were transferred to two sets of individual wells on a white plastic reaction plate (product #655075; Greiner Bio-One, Kremsmünster, Austria). In one set of wells, 200 µL free glycerol reagent were added, and 200 µL of the TAG reagent were added in the other set. Both sets were incubated at 37 °C for 1 h. One hundred microliters of each well were transferred to a clear flat bottom plate (product # 655095; Greiner Bio-One, Kremsmünster, Austria), and the absorbance was measured at 540 nm in a Tecan Infinite 200 Pro (Tecan, Männedorf, Switserland) with i-control software. Stored TAG levels were determined by subtracting the absorbance of the free glycerol from the absorbance of the total glyceride measurements.

### 2.3. Body Weight Measurement

Following eclosion, the flies were aged for 3 days, after which the male and female flies were transferred at a 1:1 number ratio to the diets for 7 days. The flies were anesthetized on a CO_2_ pad and were divided in 6 replicates per sex, per condition, with each replicate containing 5 flies. The flies were euthanized with chloroform (product #102445; Sigma-Aldrich, St. Louis, MO, USA). The wet and dry masses were measured to the nearest 0.1 mg on the Mettler Toledo XS204 scale. Flies were placed in a 37 °C oven to dry for 24 h prior to measuring the dry mass. Each replicate consisted of 5 flies, and weight data were reported as the average weight of 1 fly per replicate.

### 2.4. Fat Body Staining and Histological Analysis 

Following 7 days on the diets, the flies were anaesthetized on ice. Six females and 6 males per condition were immobilized on a dissection pad with an insect needle through the thorax. Their head, legs, wings, and ventral abdominal walls were removed with scissors. Gut, ovaries, and testes were removed to reveal the fat body. The abdomens with fat bodies were incubated in 3.7% formaldehyde for 45 min, followed by 3 washes with cold PBS. Fixed fat bodies were incubated with rhodamin phalloidin (product #P1951; Sigma-Aldrich, St. Louis, MO, USA) in PBS + 0.4% Triton X-100 (PBST) overnight at 4 °C, followed by 3 washing steps with cold PBST and an overnight incubation of the tissue with 0.125 µg/mL Bodipy 493/503 (product #D3922; Life technologies, Carlsbad, CA, USA) at 4 °C. Fat bodies were washed 3 times using cold PBST. The dorsal abdominal wall was isolated from the thorax and mounted using a fluorescence mounting medium Vectashield H1000 (Vector Laboratories, Burlingame, CA, USA). Images were taken with an Olympus Fluoview FV1000 confocal microscope and processed using ImageJ64 (Fiji) [26]. Using this software, the percentage cell area covered by lipid, lipid droplet diameter, and cell size were measured from the confocal images. To measure the percentage of the cell corresponding to the lipid area, we first measured the area of a fat body cell on merged channels images as follows: on each image, representative of one individual fat body, we selected 3 distinct fat body cells using the “free hand selection” tool, and the area of each one was measured using the “measure tool”. Afterwards, on the green channel, we measured the surface covered by lipid droplets inside each fat body cell previously selected. To this end, we manually adjusted the threshold to cover the surface of lipid droplets, and we analyzed the area and the circumference using the “analyze particles” tool. The areas of the generated objects were measured in square micrometers (μm^2^). The percentage reported was the surface covered by lipid droplets in the fat body cell relative to the surface of the cell. For these measurements, 5 abdomens (images of individual fat bodies) were analyzed per condition and sex. The data showed the average of 3 fat cells per fat body (abdomen). The lipid droplet diameter was calculated from the circumference data generated during the previous analysis, in which the ImageJ measurement tool also provided the circumference of the (lipid droplet) particles that were analyzed. To analyze the cell size, 10 fat body cells per fat body were selected using the “free hand selection” tool and measured via the measurement tool in ImageJ. Three individual fat bodies were analyzed per condition and sex. The cell size was reported in square micrometers (μm^2^). 

For histological analysis, the flies were fixed in a solution of 4% paraformaldehyde, rinsed several times in PBS and embedded in 12% gelatin to facilitate further orientation of the samples prior to sectioning. The flies in gelatin were then immersed in increasing concentrations of a sucrose solution in glycerol (5%, 10%, and 15%) for cryopreservation and frozen at –60 °C in an OCT-compound medium, using the fast-freezing device of the cryostat HM560 (Thermo Fischer Scientific). Frozen tissue sections, with a thickness of 10 µm, were placed on poly-lysine coated slides, dried at room temperature for 15 min and stored at −80 °C until staining. Staining was performed using RAL products (RAL reactifs, Martillac, France), according to the following protocol: right out of the freezer, slides were placed on a hot plate set at 60 °C for 10 min and then washed with water. Nuclear staining was performed in a Mayer’s haemalum solution for 3 min and washed in water, followed by cytoplasm staining in an eosin solution for 2 min and washing in water and then differentiation in graded ethanol baths, ending with absolute ethanol and Diasolv (Diapath, Martinengo, Italy). The slides were mounted with a Diamount medium (Diapath, Martinengo, Italy). Observations were conducted, using an Olympus IX81 microscope (Olympus Corporation, Tokyo, Japan). Pictures were taken using an Olympus XC50 camera (Olympus Corporation, Tokyo, Japan) and processed (color adjustment) using ImageJ64 (1.53n FIJI).2.5.

### 2.5. Glycogen Measurement

Glycogen content was measured using an HK assay (GAHK20; Sigma-Aldrich, St. Louis, MO, USA), in which glucose obtained from the conversion of whole-fly glycogen to glucose is measured through the enzymatic activity of amyloglycosidase [24]. Following homogenization in PBS and centrifugation at maximum speed, the supernatant was transferred to a new tube and diluted with cold PBS (*v*:*v*, 1:3). An amyloglucosidase solution (AS) was prepared by adding 3 µL amyloglucosidase (from Aspergillus niger product #A7095; Sigma-Aldrich, St. Louis, MO, USA) to 1 mL PBS. A glycogen stock solution (1 mg/mL) was prepared in PBS. The glycogen stock solution was used to prepare glycogen standards (0.16 mg/mL, 0.08 mg/mL, 0.04 mg/mL, 0.02 mg/mL, 0.01 mg/mL, and 0 mg/mL) in a mixture of PBS/AS (*v*/*v*: 50/50). Glucose standards with the same concentrations in PBS were prepared from a glucose stock (1 mg/mL) (Sigma-Aldrich, St. Louis, MO, USA). Thirty microliters of each standard were added to individual wells on a UV transparent plate. Fifteen microliters of each sample were added to two sets of wells on the same plate. Fifteen microliters of PBS and 15 µL AS were added to the first and second sets, respectively. The plate was incubated at 37 °C for 1 h. After spinning down the plate, 100 µL HK reagent was added to each well. The plate was incubated at room temperature for 15 min prior to absorbance measurement at 340 nm. Of note, Nutri-Fly^®^ Bloomington Formulation fly food contains cornmeal. Since we used whole animals for glycogen measurements, it is possible that starch in the digestive tract of the flies contributed to the overall readout in this assay and thus that the values were actually an overestimate of the actual glycogen content.

### 2.6. Ovary Staining and Measurement

Following 7 days on the diets, the female flies were anaesthetized on ice. Ovaries were dissected from 3 females per condition. The ovaries were incubated in 3.7% formaldehyde for 45 min, followed by 3 washes with cold PBS. Fixed ovaries were incubated with rhodamin phalloidin (product #P1951; Sigma-Aldrich, St. Louis, MO, USA) in PBS + 0.4% Triton X-100 (PBST) overnight at 4 °C, followed by 3 washing steps with cold PBST. The ovaries were mounted using Vectashield H1000 fluorescence-mounting medium (Vector Laboratories, Burlingame, CA, USA). Images were taken with an Olympus Fluoview FV1000 confocal microscope (Olympus corporation, Tokyo, Japan) and processed using ImageJ64 (Fiji) [26]. Using this software, the diameter of the ovaries was measured (in µm).

### 2.7. RNA Extraction and qRT-PCR

Four replicates per condition per sex, with each containing 10 flies, were collected following the 7-day diet. The samples were snap-frozen in liquid nitrogen, and fly heads were separated from the bodies. RNA was extracted from heads and bodies separately using a phenol-chloroform procedure. Tissues were homogenized in 1 mL Tri-Reagent^®^ (product #AM9738; Thermo Fisher scientific, Aalst, Belgium) and incubated for 5 min at room temperature. RNase-free chloroform was added, and following mixing and centrifugation, the upper aqueous phase was isolated and mixed with isopropanol. RNA was precipitated by centrifugation, and the pellet was washed twice with 1 mL of 75% ethanol. The pellet was air-dried and resuspended in 22 µL RNase-free water for heads and 102 µL RNase-free water for bodies. RNA was then stored at −80 °C until further processing. RNA concentrations were measured using 2 µL with a Nanodrop ND-1000 spectrophotometer (Thermo Fisher Scientific, Waltham, MA, USA). cDNA was produced using A Sensifast^TM^ cDNA synthesis kit (product# 65054; Meridian Bioscience, GC Biotech B.V., Waddinxveen, The Netherlands) in 20 µL total volume using 1 µg of total RNA, following the manufacturer’s protocol. cDNA was diluted 10-fold in nuclease-free water to a final concentration of 100 ng/µL and was stored at −20 °C until use. qRT-PCR reactions were performed on a ViiA 7 Applied Biosystems Real-Time PCR system (Thermo Fisher, Aalst, Belgium) using SYBR Green (FastGene 2x IC Green mix—low ROX (Nippon Genetics, Düren, Germany) with 500 ng cDNA template and 100 nM of each primer, in 384-well optical plates. We focused on genes involved in carbohydrate (*Pgi*, *Pepck1, Glyp*, *Tps-1*, *Zw*, and *Glys*) and fat (*Lipin*, *Fasn1*, *acetyl-CoA carboxylase* (*Acc*), *Lsd1*, *Bmm*, and *Atpcl*) metabolism and on immune-related genes (*Drs*, *Dpt*, *Relish*, *Upd3*, and *Stat92E*). We also tested three important *Drosophila* insulin-like peptides (*dIlp2*, -*3*, and -*5),* known for their involvement in metabolism regulation, as well as their possible regulators (*Dilp6*, *Eig*, *Tace*, *Upd2*, *CchA*, and *Lst*) and factors downstream of insulin signaling (*4EBP*, *IR*, and *Foxo*). We also assessed the expression of adipokinetic hormone (*Akh*)*,* important for energy homeostasis during starvation. dIlp expression levels (dIlp2, -3, and -5) were measured in the head samples. Gene expression was normalized to the housekeeping genes *Act 42A* and *LaminCa*, and all other genes analyzed in bodies were normalized to *LaminCa* and *Beta tubulin 60D*. These housekeeping genes were chosen based on the stable expression across diets and sex. Relative expression values were determined by the ∆∆CT method (Livak and Schmittgen, 2001), using the control condition (0% sucrose and 0% fat) as a reference for each sex separately. Data are displayed as heatmaps depicting the fold change in expression compared to the expression in female or male flies on the control diet (CD). All primers used were intron-spanning and are listed in Table 2. 

### 2.8. Statistics

The data were analyzed using software package Graphpad Prism 9.2.0. All data were assessed using two-way ANOVA with Dunnett’s multiple comparison, in which all conditions were compared to the CD (0S0F). For the qRT-PCR data, statistical analysis was performed on relative expression data using two-Way ANOVA with Dunnett’s multiple comparison compared to the control condition. The statistical analysis of qRT-PCR data to evaluate sexual dimorphism was conducted on ∆Ct values of males and females for each gene separately using unpaired Students *t*-test. Graphs with box plots displayed the means with the minimum and maximum values. The graph depicting the weight showes the mean with SD. For all statistical analyses, differences were considered significant, if the *p*-value was <0.05. 

## 3. Results

### 3.1. TAG Content Increases in Response to 10% Increase in Dietary Sucrose and Fat

In a dose–response experiment, we aimed to identify nutritional conditions that led to an increase in stored TAG in adult fruit flies (Appendix A). To ensure the flies survived the experimental period on the food, the 7-day survival of the flies was assessed and found comparable across all dietary conditions (Appendix A). Stored triglyceride concentrations were measured in male and female flies placed on 15 different diets at different coconut oil and sucrose concentrations (Table 1). The data from the dose–response experiment were expressed as the fold change compared to standard Nutri-Fly (condition: 0% sucrose and 0% coconut oil), hereafter referred to as “CD” per sex (Appendix A). We observed that the addition of sucrose or coconut oil did not seem to drastically change the level of TAG in female flies. In males, TAG increased in response to the addition of 5–20% sucrose and the addition of 5–20% sucrose in combination with 10% coconut oil. In contrast, diets containing 20% of coconut oil with or without sugar showed no changes in TAG. In addition, conditions containing 30% of sucrose did not increase TAG concentrations upon the addition of fat (Appendix A).

Next, we aimed to study the trends observed in the dose–response experiment in more detail by repeating the measurement of the stored TAG concentrations in a new set of experiments containing only six selected dietary conditions: the CD, the CD supplemented with 10% (FD1) or 20% fat (FD2), the CD supplemented with 10% sucrose (SD), and the CD supplemented with 10% sucrose and 10% fat (WD1) or 20% fat (WD2) (Table 1 and Figure 1). The data are expressed as an average mass (µg) of the stored TAG per fly. TAG levels were affected by diet (two-way ANOVA, F = 9.17, *p* < 0.0001) and sex (two-way ANOVA, F = 46.13, *p* < 0.0001) (Figure 1). The male flies displayed a two-fold increase in TAG concentration in conditions with 10% added sugar (SD) compared to the CD (Figure 1) and a three-fold increase in TAG concentrations in WD1 and WD2 compared to the CD (Figure 1). On the CD, females displayed a three-fold higher TAG concentration compared to males (Figure 1; N = 4, 17, and 87 vs. 50 and 47). The TAG concentrations in females increased 1.4 fold on FD2 and WD1. Since high concentrations of sugar (and fat) can cause desiccation of the flies and obscure the effects of the nutrients on fruit fly physiology, we assessed wet and dry weights. In males, the wet and dry weights did not change significantly, while wet weight was significantly decreased in females on FD2, SD, WD1, and WD2. In addition, the dry weights of females were reduced in a similar fashion in these conditions, although not significantly on WD1 and WD2. The female-specific decrease in weight led us to hypothesize that the diets might affect oogenesis. The rhodamin phalloidin staining of ovaries of the females after seven days on the diets revealed significantly smaller ovaries in females on FD2, WD1, and WD2 (Appendix A), consistent with the notion that diets supplemented with fat or a combination of fat and sugar affect oogenesis.

### 3.2. Western Diets Modify Fat Body Organization and Increase the Lipid Content of Fat Cells

The fat body is the main organ for storage of fat in fruit flies. We used rhodamin phalloidin and Bodipy staining to measure the lipid droplet area within the fat body cells of female and male flies on the different diets (Figure 2A–L). During the dissection of fat bodies of flies on western diets (WD1 and WD2), we noticed that both in males and females the fat floated out of the abdomens (not shown), suggesting an abnormal organization of the fat body. This was confirmed by histological analysis, which showed a general disorganization of the fat body and a disruption of fat cells in both males (Appendix A) and females (Appendix A). This effect was more pronounced in females, where we also found larger fat cells.

The lipid area of fat cells was affected by sex and diet (Figure 2M) (two-way ANOVA, F = 21.29, *p* < 0.0001 and F = 11.61, *p* < 0.0001, respectively). However, in the WD1 condition, the lipid area was significantly increased in both males and females compared to in the CD (Figure 2M). The mean lipid droplet diameter was significantly increased in females on all diets, except for SD, while in males lipid droplet diameter was only increased on FD1 (Figure 2O). The measurement of the cell size revealed significantly smaller cell sizes in FD1 and FD2 in females and in FD1 in males (Figure 2N). The addition of sugar or a combination of sugar and fat to the diet did not change the cell size as compared to controls (Figure 2N).

### 3.3. Glycogen Is an Important Energy Storage Molecule for Drosophila Females in Response to Dietary Sugar

Similar to in other animals, glycogen is a key storage form of carbohydrates in *Drosophila,* and its synthesis is dynamically controlled and serves as a protection mechanism against deleterious effects of high sugar concentrations [25]. We measured glycogen to determine the dynamics in glycogen metabolism in response to the diets in males and females. The females displayed a two-fold higher concentration of glycogen compared to males on the CD (Figure 3). The glycogen content was significantly lower in males and females on FD1 and FD2 (Figure 3). The supplementation of the diet with sugar alone (SD) resulted in a significantly higher glycogen content in females, while on western diets the glycogen content was lower compared to under the SD condition (Figure 3). In contrast, the addition of sugar to the diets did not increase the glycogen content in males. 

### 3.4. Expression of Carbohydrate and Lipid Metabolism Genes on Sugar and Fat-Supplemented Diets Is Sexually Dimorphic

Carbohydrate and fat metabolism depend on a series of essential enzymes that govern their breakdown or storage. We assessed the expression of key metabolic enzymes associated with glycolysis (*Pgi*), gluconeogenesis (*Pepck1)* glycogenolysis *(Glyp*), trehalose synthesis (*Tps1*), the pentose phosphate pathway (*Zw*), glycogenesis (*Glys*), lipogenesis (*Lpin*, *Fasn1*, and *Acc*), lipolysis (*Bmm*), fat storage (*Lsd1*), and TCA-cycle (*Atpcl*) (Figure 4A). Except for *Glys* (encoding glycogen synthase), all genes displayed a sexually dimorphic expression in response to the diets (Figure 4A). Indeed, *Glys* was significantly reduced in males and females in all diets (Figure 4A), suggesting reduced glycogen synthesis. Other genes showed an expression variation that was sexually dimorphic: *Glyp* (encoding glycogen phosphorylase, which is required for glycogen breakdown) mRNA was significantly increased in females exposed to FDs (Figure 4A), while in males we observed a significant increase in the expression of *Trehalose synthesis enzyme 1* (*Tps1*) on the FDs and also on the SD and WD2, although not statistically significant for the latter two (Figure 4A). 

To evaluate lipogenesis, we assessed the expression of *Acc* and *fatty acid synthase 1* (*Fasn1*) genes. Contrary to the significant (3-fold) increase in TAG levels in the males on the WDs, *Acc* was significantly reduced on WDs, while there was no significant change in the expression of *Fasn1* (Figure 4A). In the females on the WDs, where the effect on the TAG level was less pronounced, the expression of *Fasn1* was significantly induced, and *Acc* expression was not significantly changed (Figure 4A). Triglyceride breakdown in the fly is mediated by the lipase brummer (Bmm), while perilipins such as Lsd1 regulate the accessibility of triglyceride to lipases such as Bmm. While the expression of *Lsd1* was increased in females exposed to SD and WD1, *Bmm* expression was not significantly changed in females on SD and WDs (Figure 4A). Globally, the sex-dependent responses to the diets suggest that, like carbohydrate metabolism, fat metabolism is also regulated in a sexually dimorphic fashion.

### 3.5. Sexually Dimorphic Changes in the Expression of Insulin and Akh-Signaling Genes in Response to Diets Supplemented with Sugar and/or Fat

We next assessed how *dIlp2*, -*3*, and -*5* were expressed in heads of male and female adult flies following a seven-day exposure to sugar and/or fat-supplemented diets. All are important *Drosophila* insulin-like peptides with distinct metabolic roles. The addition of sugar and fat reduced the expression of *dIlp2* and *-5* in female flies, but not in males. This reduction in *dIlp*2 expression was significant in females on FD1, FD2, and WD1 (Figure 4B). In males, the addition of both sugar and fat to the diet of males significantly increased the expression of *dIlp3* (Figure 4B), while the addition of sugar or fat alone also resulted in higher *dIlp3* expression, although not significant (Figure 4B). Of note, *dIlp3* expression levels under normal conditions were significantly higher in females (Figure 4F). The addition of sugar alone significantly increased *dIlp5* expression in males (Figure 4B). These data thus showed sexually dimorphic effects on *dIlp2*, *-3*, and *-5* expression levels in response to the 7-day exposure to sugar and fat-rich diets.

Next, we analyzed the impacts of diet and sex on insulin signaling by measuring expression levels of *InR* and *Thor*, two well-known target genes of insulin signaling that encode the insulin receptor and 4E-BP, eukaryotic initiation factor 4E (eIF4E)-binding protein, which are crucial regulators downstream of the IIS and mTOR signaling pathways [28], respectively. *InR* expression depended on diet and sex (two-way ANOVA, N = 4, F = 7.659, *p* < 0.0001 and F = 9.874, *p* = 0.0034, respectively), its expression increased in males on diets containing sugar and/or fat and was significantly increased in males exposed to WD2 (Figure 4C). The transcript levels of *Thor* were dependent on diet (two-way ANOVA, N = 4, F = 5.045, *p* = 0.0013) and increased in both males and females in response to WD1 and WD2, but not significantly (Figure 4C). The expression of *Foxo*, the transcription factor that regulates *InR* and *Thor* downstream of insulin signaling, was not significantly affected by sex or diet.

Adipokinetic hormone (Akh), a glucagon homolog produced by the corpora cardiaca, mediates the mobilization of carbohydrates and fat from the fat body. Diets significantly affected *Akh* expression (two-way ANOVA, N = 4, F = 3.193, *p* = 0.0238). In female flies exposed to FD1 and WD1, the expression of *Akh* was increased, but not significantly (Figure 4C). The gene expression levels of *Thor, IR*, *Foxo*, and *Akh* was reduced in both males and females exposed to a diet supplemented with sugar alone (SD; Figure 4C). In summary, diet and sex differentially affected insulin and Akh signaling.

### 3.6. Secreted Signals That Inhibit dIlp Expression Are Upregulated in Females on FD and WD

We finally wanted to test the expression of signals deriving from the fat body and that regulate *dIlp* expression (*dIlp6* [29], Eiger [30], *Unpaired 2* [31], and CCHamide [32]) and the gut-derived signal limostatin (*Lst* [33]). Both *dIlp6* and *Eiger* (using Tace expression levels as a proxy) adipokines showed a 1.3-fold increase in =expression in females exposed to western diets as compared to females on the CD. However, this increase was only significant for the expression of *Tace* in female flies on WD2 (Figure 4D). Additionally, a trend towards an increased expression of *dIlp6* and *Eig* expression levels (1.4 and 1.2 fold, respectively) in females on FD2 as compared to the control was observed and was absent in males (Figure 4D). *Unpaired 2* was significantly increased in males on WD2 and also increased on WD1 as compared to under CD (Figure 4D). The expression of CCHa was 1.3-fold higher in females exposed to FDs and reduced in males and females placed on WD2 (Figure 4D). *Lst* expression was significantly upregulated in females exposed to FD2, and its expression was also increased in western diets (Figure 4D). 

### 3.7. Sexually Dimorphic Gene Expression in Flies under Standard Conditions 

The analysis of gene expression provided us with data of age-matched male and female flies under standard conditions (CD). We displayed the ∆Ct values in a heat map (Figure 4F) and determined statistically significant differences in ∆Ct values for each gene between males and females. Fifty percent (17/34) of the analyzed genes displayed significant sexually dimorphic expression. Five out of 12 metabolic genes were expressed in a sexually dimorphic manner. *Glys* displayed a strong female bias. Other metabolic genes with sex-biased expression are genes related to lipid metabolism, *Lsd1*, *Acc*, and *Fasn1*. These genes were significantly higher expressed in males compared to in females. Remarkably, all fat body-derived signals, except for *dIlp6*, displayed significant sexually dimorphic expression. For insulin and insulin signaling, *dIlp3* and *InR* showed higher expression in females, while *Foxo* expression was higher in males. For the immune-related genes, *Drosomycin*, *Stat92E*, and *Upd3* were sexually dimorphic.

### 3.8. Immune Response

Obesity pathogenesis in humans is associated with a chronic inflammatory state. The fat body in the fly is the central organ for nutrient storage and breakdown and is also essential in the humoral immune response to infection via the production of antimicrobial peptides (AMPs). In flies placed on sugar and/or fat-supplemented diets, the expression of Relish, a key factor in the Immune deficiency (Imd) pathway involved in the induction of the humoral immune response, was two-fold upregulated in both males and females on SD and FDs and three-fold upregulated on western diets in both sexes (Figure 4E). Additionally, *Diptericin (Dipt)*, a Relish target gene which is an AMP directed against gram negative bacterial infection [34], was increased in females on FDs, SD, and WD1, and in males on FD2 and SD (Figure 4E). In contrast, *Drosomycin* (*Drs*) expression, which is induced via the Toll signaling pathway [34], did not significantly change in response to the diets. The Jak/stat pathway with its cytokine-like ligand Upd3, which is induced upon bacterial challenge and viral infection [35], were also not induced in response to sugar and/or fat-supplemented diets in males or females (Figure 4E). In fact, the expression of *Stat92E* was significantly downregulated in flies exposed to sugar and/or fat-supplemented diets.

## 4. Discussion

Despite the frequent use of *Drosophila* for metabolic research using obesogenic diets, little is known about the differences between males and females exposed to these diets. We have defined western diets containing sugar and fat and compared the metabolic responses of males and females on western diets and other diets containing sugar or fat alone to files on the control diet. We observed that the addition of 10% sugar and 10% fat to a standard diet resulted in sexually dimorphic increases in the TAG content in male flies, showing a three-fold increase, while the TAG content in females showed a 1.4-fold increase. The increase in TAG levels in females was the result of larger lipid droplets, leading to an increased lipid area expressed as a percentage of the total area of the cell. In males, the relative lipid area was increased on the western diets, while the fat body cell size and the lipid droplet diameter did not increase, suggesting a general increase in the size of the fat body (more cells) or an increase in the number of lipid droplets per cell. So far, sex differences in lipid droplets, cell size, and number in the *Drosophila* fat body have been understudied. Our data suggested that males and females might display different fat body histologies. The significant reductions in the cell size of adipocytes in the fat body of females and males in response to different fat diets (FD1, FD2, and FD1) were remarkable. We hypothesize that flies reared on fat diets consumed relatively less protein, a situation similar to a yeast protein-poor diet for which a small decrease in adipocyte size has been reported following 24 h on a yeast-poor diet [36]. In addition, the size of adipocytes in response to diets could also depend on the timing of the observation. In rats challenged with high fat diet, new, smaller adipocytes arise after three days on the diet [37].

Our data indicated that glycogen metabolism is also regulated in a sexually dimorphic manner, with female flies increasing the glycogen content in response to supplementation with sugar in the diets, something not observed in males. It is possible that females are more readily capable to store excess sugar as glycogen rather than fat, with a resultant more modest increase in TAG levels in females as compared to in males. This, however, remains to be tested. An alternative explanation for the limited increase in TAG levels in females in response to the diets can be the reduced number of mature oocytes in the ovaries of these flies. It has been shown that oocytes store lipids [38] and glycogen [39,40]. Since females on the CD displayed fully developed ovaries with mature oocytes while the female flies on FDs and western diets displayed small ovaries, we believe that the relative increase of TAG levels measured in the females on these diets could actually be an underestimate of the actual increase of TAG levels in the fat body of those females. In parallel, the apparent lower glycogen levels in female flies on the fatty diets may also be due to the reduced size of ovaries and fewer mature eggs in the flies on the fat diets as compared to those on the CD. However, while females on the western diets also displayed smaller ovaries with fewer oocytes, the glycogen levels in these flies were not significantly different compared to those of females on the CD, suggesting that in these conditions there was an increase in the glycogen content in the fat body. Nevertheless, a part of the sexually dimorphic response in lipid and glycogen levels on these diets is likely due to reduced oogenesis, something that could be experimentally validated using female flies with arrested oogenesis as described in [41].

Despite these issues, we observed sexually dimorphic regulation of lipid and carbohydrate homeostasis, which is consistent with the sex-biased expression of genes associated with lipid and carbohydrate metabolism that we observed and has been reported previously [18,42]. In our study, half of the genes analyzed were expressed in a sexually dimorphic fashion in adult flies under standard conditions, suggesting an inherent sex difference in expression. The data indicated that the sexual dimorphism in the responses to the diets is governed at two levels, with the first at the level of the metabolism of carbohydrates and lipids and the second at the level of the regulatory signals and pathways that govern metabolic processes. At the first level, the expression of key metabolic genes suggested that female flies increased lipogenesis and downregulated genes for lipolysis. Whereas the expression of genes from a specific lipid-associated process was not specifically increased in males, we observed an increased expression of *trehalose synthesis enzyme 1* (*Tps*), suggesting that trehalose metabolism plays an important role in males in the protection against metabolic stress. In a study analyzing transcriptome differences in heads and bodies of males and females with a normal and HFD, a similar observation was made where female flies increased the expression of genes associated with lipid metabolism while males did not [18]. In bodies of males on HFD, the transcriptome analysis revealed an enrichment of genes associated with oligosaccharide synthesis, consistent with the male-specific increase of *Tps* on fat and western diets in our experiments [18]. Since we only analyzed the mRNA expression levels of key metabolic enzymes, we cannot exclude further sexually dimorphic differences at the protein levels of these enzymes, as allosteric regulation is known to be common for many metabolic enzymes [27].

Insulin and Akh signaling are important regulators of metabolism. The insulin-producing cells (IPCs) in the pars intercerebralis of the brain are the main source of three important *Drosophila* insulin-like peptides, *dIlp2*, *-3* and *-5*, while dIlp3 and dIlp5 are also produced by muscle cells of the midgut, ovarian follicle cells, and principal cells of the Malpighian tubules, respectively [43]. It is suggested that these *dIlps* have distinct metabolic roles, with *dIlp2* regulating glucose metabolism, *dIlp3* regulating lipid metabolism, and *dIlp5* regulating the response to amino acids [44]. We observed a significant downregulation of *dIlp2* and *dIlp5* in the heads of females on fat and western diets. We hypothesized that the increased expression of regulatory signals, *dIlp6*, *Eiger* (and by extension *Tace*), and *Limostatin* deriving from the fat body and gut in females on these diets, induced a strong inhibitory effect on the expression of *dIlp2* and *-5*. Previous studies observed a positive correlation between *dIlp5* gene expression and the body size in females. This correlates with our observation of the reduced *dIlp5* expression and the reduced body weight of females on the diets [45]. In contrast, *dIlp3* expression, which had a female-biased expression under normal conditions, increased significantly in males on fat and western diets. This would be consistent with the increased expression of the *InR* gene in males in these conditions [46]. While dIlp3 was previously indirectly linked to lipid metabolism [47], recent studies show that dIlp3 might play a role in regulating trehalose concentration in hemolymph [48], which would correspond to the observed male specific increase in *Tps*. However, the reason for the strong male-specific induction of *dIlp3* expression on these diets remains to be elucidated. Other studies confirm the sexually dimorphic expression of some of the genes we have studied [41,49] and unravel their roles in the sexually dimorphic regulation of lipid homeostasis. We observed a higher expression of the *Akh* gene in males under standard conditions. This is in accordance with the observation from a recent study by Wat et al., showing that Akh neurons are modulated by the sex determination gene *transformer Tra*, lowering Akh levels in females and increasing lipid storage, thereby regulating the sex-specific differences in whole-body fat storage [40].

We observed important differences between male and female *Drosophila* lipid and carbohydrate metabolism and their regulation. Since excess sugar can be stored as glycogen, it is possible that obesogenic diets based on high sugar (HSDs) induce obesity to a lesser degree than HFDs, because sugar can be shunted to glycogen rather than triglycerides. Therefore, we propose that a combination of sugar and fat in the western diets will exacerbate the obesity phenotype and lipid dysregulation. Indeed, while HSDs and HFDs can induce similar effects such as the increase in the TAG content and insulin resistance, the gene expression analysis on the heads of female flies on either diet revealed notable differencs on gene expression with distinct metabolic genes expressed for each diet and the enrichment of genes associated with immune function on HFDs and genes relating to checkpoint kinases in HSD [50]. We propose that a combination of sugar and fat in the diet induces different kinds of stress and thus affects more processes than either diet alone, making it more representative for human obesogenic studies. Studies using western diets as obesogenic diets are scarce. One study challenged male and female wild-caught *Drosophila simulans* with a diet supplemented with sugar, fat, and salt and evaluated the sexually dimorphic effect on fly activity, food preference, and oviposition preference [19]. The authors observed that the western diet strongly reduced lifespan, locomotor function, sleep, reproduction, and mitochondrial function and that exercise can partially rescue lethality and locomotor function. The observed decreased longevity of flies on the western diet could, however, be the result of osmotic stress, since the addition of salt would increase the osmolarity of the food [19]. To ascertain that our observations in flies on western diets were the result of metabolic challenges, rather than osmotic challenges [14], we showed that although female flies did lose weight on fat, sugar, and western diets compared to female flies on the CD, the differences between wet and dry weight were not significantly different in flies on the diets compared to under control conditions. These data indicated that a modest increase in sugar (10%) in combination with a modest (10%) increase of fat increase the TAG concentrations in adult fruit flies, without causing desiccation or osmotic stress to the flies. We proposed that the weight reduction in females is related to the observed reduction in the ovarian size and the reduced oogenesis, since obesity and insulin resistance induced by HSDs and HFDs are described to disrupt ovarian metabolism and size [8,51].

Obesogenic diets also induce changes in immune processes. In humans, obesity is associated with immune dysregulation in adipose tissue, resulting in low-grade inflammation and increased risk for development of metabolic diseases such as type 2 diabetes [52,53]. In *Drosophila*, HFDs and HSDs are also associated with immune responses [35,54]. In a study assessing the immune response to HFDs, an upregulation of *Upd3* expression and signaling through the JAK-STAT pathway was observed [35], while in our study, *Upd3* expression was not significantly increased and even slightly reduced in females. We observed a significant increase in the expression of *Relish* in males and females exposed to western diets. Relish is a *Drosophila* NF-κB protein in the Imd pathway, homologous to the TNF pathway in humans [55]. The importance of the Imd pathway in metabolism has been previously reported, showing that the persistent activation of the Imd pathway in the fat body results in hyperglycemia, depleted fat reserves, and developmental delay [56]. Relish itself also plays a role in the lipid metabolism restraining fasting-induced lipolysis, involving proteins such as Brummer, to conserve trigyceride levels during metabolic adaptation [57]. These data suggest that comparable to mammals, metabolic stress induced by obesogenic diets induce changes in immune functions.

## 5. Conclusions

In this study, we defined an obesogenic diet based on the addition of sucrose and coconut oil to a standard fruit fly diet. This study is the first to demonstrate that the exposure of adult *Drosophila* males and females to western diets induces sexually dimorphic responses in metabolism and gene expression.

## Figures and Tables

**Figure 1 biomolecules-12-00033-f001:**
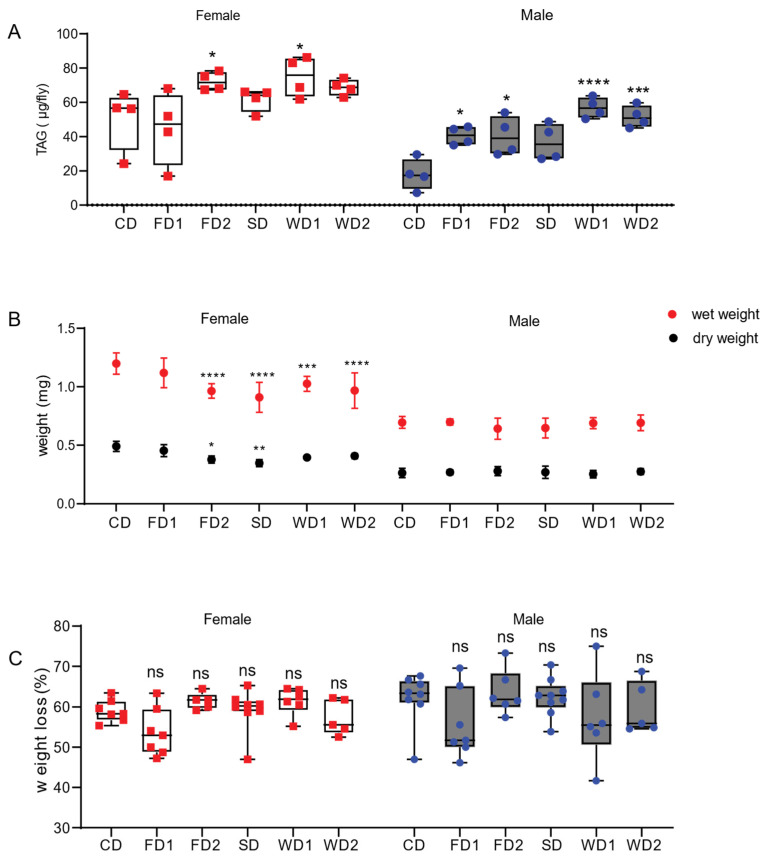
TAG levels and body weights in adult male and female flies exposed to diets supplemented with sugar, fat, or a combination of both. (**A**). TAG concentrations in adult male and female *Dahomey* flies exposed to normal diet (CD), fat (coconut oil)-supplemented diet (FD1 and FD2), (sucrose) sugar-supplemented diet (SD), and western diets (WD1 and WD2). TAG levels were expressed as µg TAG per fly in females and males. Box plots display the medians, and whisker plots indicate the minimum and maximum values. Western diets significantly increased TAG levels in males and females. *p*-values were determined using two-way ANOVA. Main effects: diet, *p* < 0.0001; sex, *p* < 0.0001 and their interaction; ns, with a Dunnett’s multiple comparisons test comparing the data on the diets to CD per sex. * *p* < 0.05, *** *p* < 0.0005, **** *p* < 0.0001. (**B**) Mean wet and dry weights of adult male and female *Dahomey* flies exposed to the diets. The body weights of 5 flies were measured in at least 5 replicates per sex per condition. The weight per fly was calculated per replicate. The data are shown as the mean and the SD. The fat and/or sugar-supplemented diets reduced the body weights of females, but not those of males. Statistical analysis was conducted using two-way ANOVA for wet weight data and dry weight data separately. For the wet weight, main effects were: diet, *p* < 0.0001; sex, *p* < 0.0001 and their interaction; *p* = 0.0002. For the dry weight, main effects were: diet, *p* < 0.0001; sex, *p* < 0.0001 and their interaction; *p* < 0.0001. *p*-values were calculated using Dunnett’s multiple comparisons test comparing the data on the diets to those on the CD (0S0F) per sex. * *p* < 0.05, ** *p* < 0.005, *** *p* < 0.0005, **** *p* < 0.0001. (**C**) The percentage of body weight that was lost after dehydration (dry weight). The percentage of lost bodyweight was not significantly different in the flies on the diets as compared to those on the CD. The data are shown as box and whiskers plots displaying the median, minimum, and maximum values. *p*-values were determined using two-way ANOVA. Main effects: diet, *p* = 0.03; sex, ns and their interaction; ns, with a Dunnett’s multiple comparisons test comparing the data on the diets to those on the CD (0S0F) per sex.

**Figure 2 biomolecules-12-00033-f002:**
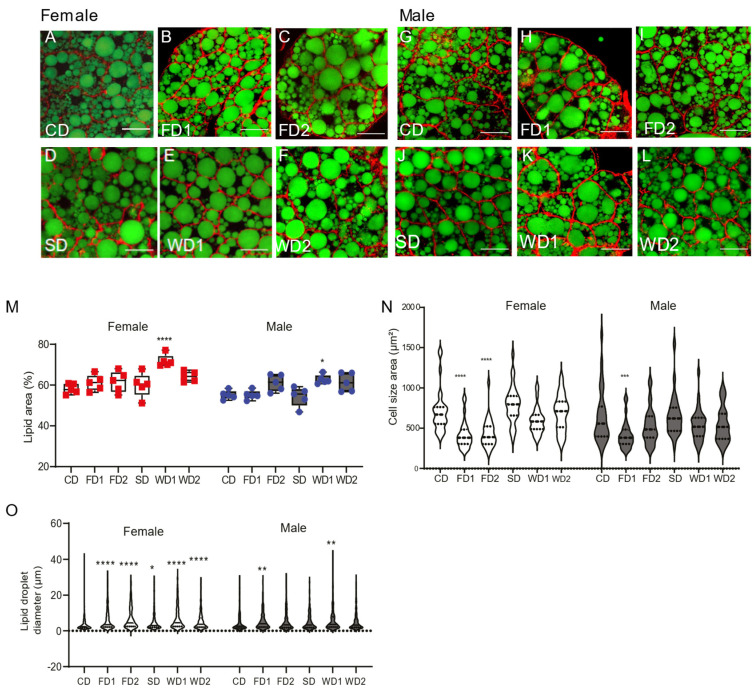
Changes in lipid area, cell size, and lipid droplet diameter in female and male flies on the diets supplemented with sugar, fat, or a combination of both. (**A**–**L**) Staining of lipid droplets with Bodipy (Green) and cytoskeleton with rhodamine phalloidin staining (red) on the abdomens of female (**A**–**F**) and male (**G**–**L**) *Dahomey* flies exposed to the normal diet (CD), fat (coconut oil)-supplemented diet (FD1 and FD2), (sucrose) sugar-supplemented diet (SD), and western diets (WD1 and WD2). Scale bars represent 20 µm. (**M**) Quantification of the percentage lipid area relative to the cell area. The percentage lipid area was increased on western diets in both females and males. Data are shown as boxplots displaying the median, minimum, and maximum values. *p*-values were determined using two-way ANOVA. Main effects: diet, *p* < 0.0001; sex, *p* < 0.0001 and their interaction; ns, Dunnett’s multiple comparisons test comparing the data on the diets to those on the CD per sex. *** *p* < 0.0005, **** *p* < 0.0001. (**N**) Quantification of the cell size of fat bodies in females and males exposed to different diets. Data are shown as violin plots displaying the median and quartiles. *p*-values were determined using two-way ANOVA. Main effects: diet, *p* < 0.0001; sex, *p* = 0.03 and their interaction; *p* = 0.048 with Dunnett’s multiple comparisons test comparing the data on the diets to those on the CD per sex. *** *p* < 0.0005, **** *p* < 0.0001. (**O**) Quantification of the lipid droplet diameters in females and males exposed to different diets. Data are shown as violin plots displaying the median and quartiles. The two-way ANOVA main effects are: diet, *p* < 0.0001; sex, ns and their interaction; *p* < 0.0001 with a Dunnett’s multiple comparisons test comparing the data on the diets to those on the CD per sex. * *p* < 0.05, ** *p* < 0.005, *** *p* < 0.0005, **** *p* < 0.0001.

**Figure 3 biomolecules-12-00033-f003:**
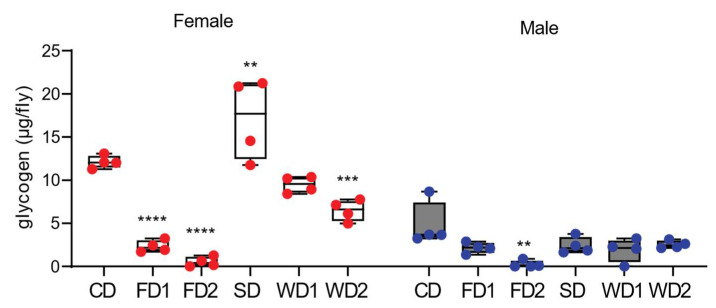
Glycogen levels in adult male and female flies exposed to diets supplemented with sugar, fat, or a combination of sugar and fat. The glycogen levels in adult male and female *Dahomey* flies exposed to the normal diet (CD), fat (coconut oil)-supplemented diet (FD1 and FD2), (sucrose) sugar-supplemented diet (SD), and western diets (WD1 and WD2). Glycogen levels are expressed as µg per fly in females and males. Box plots display the median values, and whisker plots indicate the minimum and maximum values. The diets supplemented with fat decreased the glycogen contents in females and males. The addition of sugar to the diet increased the glycogen content only in females. *p*-values were determined using two-way ANOVA main effects as following: diet, *p* < 0.0001; sex, *p* < 0.0001 and their interaction; *p* < 0.0001 with a Dunnett’s multiple comparison test comparing the data on the diets to those on the CD per sex. ** *p* < 0.005, *** *p* < 0.0005, **** *p* < 0.0001.

**Figure 4 biomolecules-12-00033-f004:**
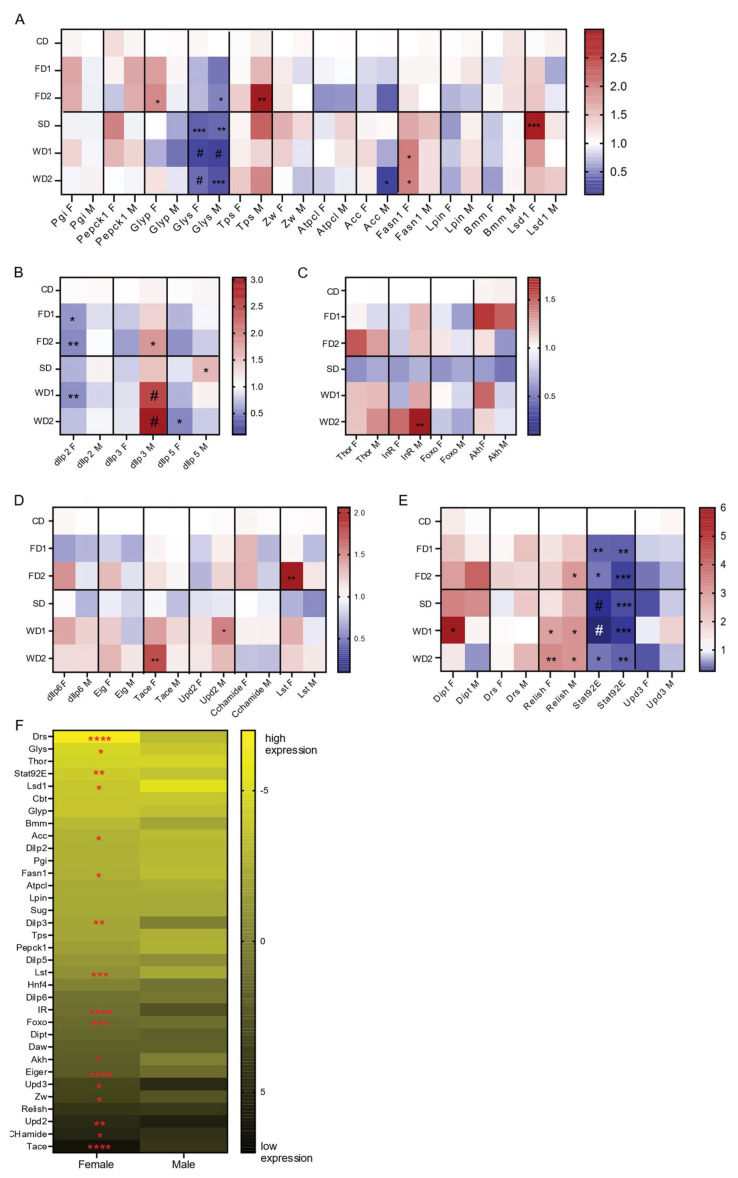
qRT-PCR analysis of gene expression in the heads and bodies of adult male and female flies exposed to diets supplemented with sugar, fat, or a combination of both. Data are depicted as heat plots displaying the fold change in the gene expression relative to the CD per sex. Statistical analysis with two-way ANOVA were conducted followed by Dunnett’s multiple comparison post-hoc test: * *p* < 0.05, ** *p* < 0.005, *** *p* < 0.0005, # *p* < 0.0001. Gene expressions are shown: (**A**) key metabolic genes (Two-way ANOVA main effects: diet, ns; sex, *p* < 0.0001 and their interaction; *p* < 0.0001); (**B**) *dIlp* genes (two-way ANOVA main effects as following: diet, *p* = 0.0016; sex, *p* < 0,0001 and their interaction; *p* = 0.0028); (**C**) insulin signaling pathway and adipokinetic hormone (*Akh*); two-way ANOVA main effects: diet, *p* < 0.0001; sex: *p* = 0.001 and their interaction; ns); (**D**) fat body and gut-derived signals that regulated insulin production and release (two-way ANOVA main effects: diet, *p* < 0.0001; sex, *p* < 0.0001 and their interaction; *p* = 0.0009); (**E**) immune-associated genes (two-way ANOVA main effects: diet, ns; sex, *p* = 0.03 and their interaction; ns); (**F**) ∆Ct values for male and female flies on the standard diet (CD). *p*-values were determined by Students *t*-test. * *p* < 0.05, ** *p* < 0.005, *** *p* < 0.0005, **** *p* < 0.0001.

**Table 1 biomolecules-12-00033-t001:** Materials and methods.

Name	Sugar (%)	Fat (%)
**Standard**	0	0
	5	0
**SD**	10	0
	20	0
	30	0
**FD1**	0	10
	5	10
**WD1**	10	10
	20	10
	30	10
**FD2**	0	20
	5	20
**WD2**	10	20
	20	20
	30	20

**Table 2 biomolecules-12-00033-t002:** Primer sequences for qRT-PCR.

Name	Forward 5′–3′	Reverse 5′–3′
*Acc*	GGCTATGCTGCGCTTAACA	GCCTCTGTTTTGTGGGTGAC
*Act42*	TGCAAAAGGAAATCACGGCG	CCGCCGATCCAAACAGAGTA
*Akh*	AGACCTCCAACGAAATGCTG	GTGCTTGCAGTCCAGAAAGAG
*Atpcl*	CTTCTGACCATCGGGGATCG	CAGGTTGGTGTCGTATGCCT
*Beta tub 60d*	CAAATCGGCGCTAAGTTCTGG	CCCACGTAGATGCCATTGCT
*Bmm*	AATGGCGTCGAATCAGACTT	AACACAGATGGGGATTTGGA
*Cbt*	ATGCCTTCTCGCTCTCATGT	TCCTGGAAAGAAGTGGCATC
*Daw*	CAAGCGGGAGTACTATGCCC	CCGGGATGGTTGTAGCTGAG
*dIlp2*	AGCAAGCCTTTGTCCTTCATCTC	ACACCATACTCAGCACCTCGT
*dIlp3*	GCAATGACCAAGAGAACTTTGGA	GCAGGGAACGGTCTTCGA
*dIlp5*	GCTCCGAATCTCACCACATGAA	GGAAAAGGAACACGATTTGCG
*dIlp6*	TGCTAGTCCTGGCCACCTTGTTCG	GCTTCCCGAAACTGTTGGGAAATAC
*Dipt*	TACCCACTCAATCTTCAGGGAG	TGGTCCACACCTTCTGGTGA
*Drs*	ACCAAGCTCCGTGAGAACCTT	TTGTATCTTCCGGACAGGCAG
*Eiger*	TCCTAGTCCGCAAAGGTGAA	CAAGTGGAAATGGGCTGCTG
*FASN1*	GTTGGGAGCGTGGTCTGTAT	GCACACCGAAGAACTGTTGG
*Fit*	TCGTTGGTCTGAGGAGGACA	CCAGTTGACAGAGTGCGGAT
*Foxo*	TTGGAAGATAATAACTGCGCCTCT	AAATTCGTCTATCGGCTGCG
*Gadph*	CTCGACTCACGGTCGTTTCA	GGTGATCTTCTGGCCGTTCA
*Glyp*	ATACAACAACAACCACGTAAACAC	GGATGTAGTCACCATCGTTGAAG
*Glys*	AGTCCTACTTCATCGCGGCA	GTCTCCTCATCCACGGAGTC
*HexC*	GCGGAGGTGCGAGAACTTAT	AGCGACTGTACACTTCCTGC
*Hnf4*	GGCGACGGGCAAACATTATG	CGCAAATCTGCAAGTGTACTGAT
*InR*	AAGCGTGGGAAAATTAAGATGGA	GGCTGTCAACTGCTTCTACTG
*LaminC*	TCCACCCAACAATCTGGTGAT	GCAACATCCTCTTTGTCGGC
*Lpin*	ATCCCACGTCCCTGATATCG	TTCATCTTGGTTGGTTAGC AGG
*Lsd-1*	TCACAATCTCACGGCTGGAC	GGCTACCATAGAACGCCAGC
*Lst*	TTTGCGTACACTTGGCAACTGG	TGAAGCCATTGTCGTCGTCAT
*Pepck1*	GCCTGAGCTCATTGAACAAAG	ACCGCAATTGTCCTGGCGCA
*Pgi*	GCATTCCAAGGAGCCTGAGT	ATGTTGGTGCTGTCAATGCC
*Rel*	TTCCGTGAAAAGCTGACCCG	GAGTGGGCTGGTGAAAAGGA
*Stat92E*	CAGGGCGTTCTCTGACCATT	TTCGGGCTCTTGACGCTTTG
*Sug*	CCAGTCCGTGATCATGAAGGCTC	GTGCCAGTGCATCCAAGGTGTCG
*Tace*	AGAATTGCGCGGCCCTG	CTTGGCACCCCTTTTCACCA
*Thor*	CATAGCAGCCACACAAGCTC	GGTGAAGCGGACATCTTAGC
*Tps1*	TTCCTGCACATCCCATTCCC	TCACAACCCAACATACCCTGT
*Upd3*	ACTGGGAGAACACCTGCAAT	GCCCGTTTGGTTCTGTAGAT

## Data Availability

All data are included in the manuscript and Appendix A.

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
