# Peer review of "Sexual Dimorphism in Metabolic Responses to Western Diet in Drosophila melanogaster"

_biomolecules, 2021, doi:10.3390/biom12010033_

Round 1

Reviewer 1 Report

The study by De Groef describes sex differences in the metabolic response to several dietary treatments in D. melanogaster. The authors analyzed fat and glycogen content and expression of several genes involved in energy homeostasis in males and females to illustrate differences in their response to obesogenic diets. The study is in general interesting, and the manuscript is very well written. I have nevertheless some comments, concerns and questions, listed below:

  1. What is the DaHomey strain? Do the authors perhaps mean the wild-type strain Dahomey, or the commonly used whiteDahomey line created in the Linda Partridge lab?
  2. Females deposit fat and glycogen in the growing oocytes. Thus, the observed differences in the total energy reserves are likely caused not only by differences in the stores in the fat body but also by changes in the oogenesis. To fully resolve this issue, one would have to analyze sterile females with oogenesis arrested prior to the deposition of energy reserves into the oocytes. I do not insist on adding these experiments but would highly recommend acknowledging and discussing these issues in the manuscript.
  3. Several recent papers analyzed sex-specific differences in energy metabolism in Drosophila. Some even added experiments where they tested the roles of the sex determination genes by including the standard feminization experiments by manipulations of Transformer. Adding this type of experiment would help to differentiate the role of the metabolic effects caused by the sex-specific differences in the nervous system from the impact of oogenesis. I do not insist on adding these experiments but I would highly recommend acknowledging these issues in the manuscript. 
  4. Please add more information on rearing the flies on the fat diet. Did you keep the vials in a horizontal position, or was this not necessary to prevent enrichment of the upper layers of the food by oil? Did you use filter paper?
  5. The fly food recipe does not seem to contain methylparaben or any other preservative to prevent bacterial growth (just propionic acid to inhibit moulds). Is this correct?
  6. The fly food contains a significant proportion of starch (soy flour, cornmeal). Unfortunately, this is not the best recipe when one intends to measure glycogen content. In the amyloglucosidase assay, starch is degraded to sugar, which leads to the over-estimation of the glycogen content. Thus, part of the observed differences in the glycogen reserves is likely caused by differential food intake or storage in the gut. You should at least acknowledge this issue somewhere in the text.
  7. Although I routinely use the TAG assay, I am afraid I do not fully understand the description of the method here. What is the rationale to include the step when 25 ul of the homogenate is added to a white plastic reaction plate for incubation and only then transferred to the transparent plate for the absorbance measurement? Why not transfer the homogenate directly to the transparent plate already for the incubation step?
  8. In the TAG assay, the authors changed the original protocol by Hildebrandt et al, using only 220 µl of the crushing buffer. Why? Could you provide some tests to show that the assay is still working in a linear range and you still get all the fat into the solution?
  9. Please add individual data points to all bar graphs and box plots (including the supplement) for the sake of consistency.
  10. Fig S1B – please add a bar graph where survival after the 7 days will be depicted as a proportion and analyzed adequately. The Kaplan Meier curve is great, but adding a bar graph would help to visualize the potential differences even better.
  11. Which part of the ovary can be seen in Fig S2D?
  12. Insulin-like peptides 2,3 and 5 are also expressed outside the brain, and, at least in the case of Dilp5, there is a sexual dimorphism already in the expression pattern. So please, at least discuss the role of potential changes in the Dilps expression outside of the head tissue and its significance for the measured data.

Reviewer 2 Report

Summary: This study describes experiments aimed at understanding the interactions of sex and diet on metabolic parameters and gene expression using the Drosophila melanogaster model system.  The authors do a nice job titrating in different levels of sugar and fat into the diet and measuring the storage of triglycerides and glycogen as well as the expression of several metabolic and immune genes and comparing these between male and female flies.  The Results are shown clearly, and I was pleased with the level of detail in the Methods section and the description of the statistical tests used throughout this study.  However, I do have a few comments that I would like the authors to address.

Major Comments

  1. In the Results section describing Figure 4, the authors mention a number of increases or decreases in expression that aren’t indicated as statistically significant on the figure (i.e., lines 391-392 – Bmm expression is decreased in females fed SD and WD diets). The authors need to be very careful about describing changes in expression that are not statistically significant as they are over-interpreting their results (unless I am missing something with how the heat maps are depicted; if that is the case, then some more clarity needs to be included in the description of these figures).
  2. The results in Figure 4F showing sexually dimorphic expression of many of the genes examined in this study is a very interesting result and it seems to be getting lost amid the descriptions of the effects of the different diets on the expression of these genes. Perhaps it might make sense to include a separate section in the Results just describing the sexually dimorphic changes in gene expression on the standard diet.
  3. It is interesting that feeding flies coconut oil decreases fat body cell size. The authors should speculate as to why this may be in the Discussion section of the manuscript.
  4. One of the major limitations of this study is that the authors are examining the expression of a number of genes coding for metabolic enzymes (in Figure 4) and not the activity of the enzymes themselves (many metabolic enzymes are regulated allosterically and not transcriptionally). The authors should spend some time in the Discussion describing this limitation and how it may alter the interpretation of their gene expression results.  With that said, the authors do present a lot of gene expression data that could be connected.  For example, if there are alterations in expression of the dIlp genes, this could alter the activity of insulin signaling in the fly and change the expression of insulin signaling target genes like InR and 4EBP.  Some of the results from Figures 4B and 4C are consistent like this (i.e., dIlp3 expression is increased in WD males and InR expression is also increased in WD males), but some results are not consistent (i.e., dIlp2 expression is decreased in WD females, but there is no effect on InR expression in WD females).  Perhaps this could be due to changes in dIlp secretion?  An expanded discussion of the gene expression results and how they connect together will help put these sometimes disparate results into context.
  5. An interesting result from this study is that females store more glycogen than males on sugar diets. Allan Spradling’s group (Sieber et al., 2016) has shown a role for glycogen metabolism in the fly ovary and these results might be useful to include in the Discussion to help put the data from Figure 3 into context.

Minor Comments

  1. In Figures 1 and 3, I think it would be nice to include the diet labels (i.e., SD, FD1, WD2, etc.) underneath each condition on the graph. I found that while I was reading the text and the different diets were being referred to by these labels, I didn’t always remember which diet had which percentages of sucrose, coconut oil, etc. so it took me a second to figure out which part of the graph I should be looking at.  I think adding these labels would increase the clarity of these figures.
  2. In lines 327-328, the graph depicting cell size is indicated as panel O while the LD diameter is indicated as panel N and these panels are reversed in the figure.
  3. In line 329, the authors say that the addition of sugar increases cell size, but this condition is not indicated as statistically significant in the figure. If this is not statistically significant, the authors shouldn’t state that there is an increase in cell size here.

Round 2

Reviewer 1 Report

The authors have answered all my comments and questions. It was a pleasure to review this manuscript. I wish the authors all the best in future research. 

Author Response

We thank the reviewer for his or her input, the review process has provided us with new insights that will direct our future research.
We are happy to read that the reviewer enjoyed our work.

Kind Regards

Sofie De Groef

Reviewer 2 Report

I thank the authors for addressing all of my comments.  The updated labels on the Figures make them much clearer and easier to follow and the updated text provides appropriate interpretation and context of the results presented.  I only have two minor comments:

1)In Figure S2B, there are asterisks that I assume are indicating statistical significance.  However, in the figure legend, it states that a One way ANOVA was performed and a p value is given, but it doesn't state whether post-hoc tests were performed like in the other figures.  Do these asterisks indicate statistical significance from a post-hoc test?  What the asterisks indicate needs to be included in the Figure legend.

2)In Figure 2O, there is a single asterisk above the SD condition in females.  What does this asterisk refer to?  There is no indication of a single asterisk in the Figure legend and the Results text states that there is no difference with this condition compared to the control condition.  Clarification (or removing the asterisk if it is there in error) needs to be included here.

Author Response

We thank the reviewer for his or her remarks and his or her thorough reading our manuscript. We have adjusted the legends of the figures mentioned by the reviewer. In fact, in FigS2b, a significant indication was missing, this was adjusted in the figure.. Below I provide a detailed response to the reviewer.

I thank the authors for addressing all of my comments.  The updated labels on the Figures make them much clearer and easier to follow and the updated text provides appropriate interpretation and context of the results presented.  I only have two minor comments:

1)In Figure S2B, there are asterisks that I assume are indicating statistical significance.  However, in the figure legend, it states that a One way ANOVA was performed and a p value is given, but it doesn't state whether post-hoc tests were performed like in the other figures.  Do these asterisks indicate statistical significance from a post-hoc test?  What the asterisks indicate needs to be included in the Figure legend.

We thank the reviewer for this remark. Indeed, the post hoc performed on these data was a Dunnett's multiple comparisons, comparing the means of the different dietary conditions to the control diet. The legend was adjusted in line 332-334 "One way ANOVA, F = 4.815, p = 0.0025, with a Dunnett’s multiple comparisons test comparing the data on the diets to the control diet (CD) per sex. * p<0.05, ** p<0.005"

2)In Figure 2O, there is a single asterisk above the SD condition in females.  What does this asterisk refer to?  There is no indication of a single asterisk in the Figure legend and the Results text states that there is no difference with this condition compared to the control condition.  Clarification (or removing the asterisk if it is there in error) needs to be included here.

We thank the reviewer for this remark. Indeed, the data indicate (post hoc dunnett's multiple comparisons) that the lipid droplet diameter is slightly, but significantly (at the level p<0.05 (p=0.045)) in the Sugar diet condition as compared to the lipid droplet diameter in the control condition.  The legend was adapted to include information regarding the * p-level, line 364